# The optimum parameters and neuroimaging mechanism of repetitive transcranial magnetic stimulation to post-stroke cognitive impairment, a protocol of an orthogonally-designed randomized controlled trial

**Ling-Xin Li**[1,2], **Jing-Kang Lu**[1☯], **Bao-Jin Li**[1☯], **Qiang Gao**[1,2], **Cheng-Qi He**[1,2]*, **Shi-Hong Zhang**[3], **You-Jin Zhao**[4], **Shuai He**[5], **Qian Wen**[6☯]

**1** Department of Rehabilitation Medicine, West China Hospital, Sichuan University, Chengdu, Sichuan, China, **2** Key Laboratory of Rehabilitation Medicine in Sichuan Province, West China Hospital, Sichuan University, Chengdu, Sichuan, China, **3** Department of Neurology, West China Hospital, Sichuan University, Chengdu, Sichuan, China, **4** Huaxi MR Research Center (HMRRC), Department of Radiology, West China Hospital of Sichuan University, Chengdu, Sichuan, China, **5** Department of Radiology, West China Hospital, Sichuan University, Chengdu, Sichuan, China, **6** Department of Integrated Traditional and Western Medicine, West China Hospital, Sichuan University, Chengdu, Sichuan, China

☯ These authors contributed equally to this work.
* hxkfhcq2015@126.com

# Abstract

## Objective

Repetitive Transcranial Magnetic Stimulation (rTMS) has been used in cognition impairment due to various neuropsychiatric disorders. However, its optimum parameters and the neuro-imaging mechanism are still of uncertainty. In order to simulate a study setting as close to real world as possible, the present study introduces a new orthogonally-designed protocol, consisting of the rTMS intervention with four key parameters (stimulating site, frequency, intensity and pulse number) and three different levels in each one, and aims to investigate the optimum parameters and the brain activity and connectivity in default mode network (DMN), dorsal attention network (DAN), central executive network (CEN) following rTMS intervention to post-stroke cognition impairment (PSCI).

## Methods

A single-center, orthogonally-designed, triple-blind randomized controlled trial will be conducted and forty-five PSCI patients will be recruited and randomly assigned to one of nine active rTMS groups based on four rTMS paraments: stimulating site, frequency, intensity and pulse number. Neuropsychological, activities of daily living, quality of life and functional magnetic resonance imaging (fMRI) evaluations were be performed pre-, post- and 3 months after rTMS.

**Data Availability Statement:** All relevant data from this study will be made available upon study completion.

**Funding:** This work was supported by the Post-Doctor Research Project, West China Hospital, Sichuan University (No. 2020HXBH018); the Chengdu Science and Technology Bureau (No. 2019-YF05-00061-SN); and the Health Department of Sichuan Province (No. S15063). The funders played no role in the design, conduct, or reporting of this study.

**Competing interests:** The authors have declared that no competing interests exist.

## Discussion

This study evaluates the optimum parameters of rTMS for patients with post-stroke cognition impairment and explores the alteration of neural function in DMN, DAN, CEN brain network. These results would facilitate the standardized application of rTMS in cognition impairment rehabilitation.

## Introduction

Post-stroke cognitive impairment (PSCI) is one of the vascular cognitive impairments and probably develop into dementia [1]. It was reported that 20–70% stroke survivor suffered from the cognitive impairment within 6 months after stroke attack [2, 3]. It is not only a common, important disability after stroke but also could hinder the rehabilitation of stroke-related sequelae including motor dysfunction, sensor deficits and the limited activities of daily living [4]. The pathogenesis of PSCI is multifactorial being similar with VCI. In terms of brain structure, it is generally related to focal, multifocal or diffuse cortical and/or subcortical large infarct, microinfarcts or lacunes, as which is often affected important brain areas (thalamus, frontobasal and/or limbic systems, etc.) [5]. Functionally, the PSCI is due to the disfunction of neuronal metabolism, activation and connectivity, while the exact mechanism between vascular lesion, cognition, and neuroplasticity are not completely understood [6]. Current research shows that many non-invasive neuromodulatory techniques could improve cognitive impairment via neuroplasticity [7]. As a non-invasive brain stimulation,the repetitive transcranial magnetic stimulation (rTMS) was widely used as a primary or an add-on treatment to improve cognitive function in patients with stroke [8]. Several meta-analyses confirmed its promising and positive effect on PSCI [9–11]. In these studies, mostly rTMS schemes included four primary parameters: stimulating site, frequency, intensity and pulse number, however, the rTMS intervention scheme varied substantially and the considerable heterogeneities of rTMS parameters were the great challenging in its clinical practice. Furthermore, although accumulating studies have shown that rTMS could improve the cognitive function in many neurological and psychological diseases [12, 13], the brain functional mechanism is still unclear. The present study hypothesized that rTMS could promote the cognition recovery with an optimal parameter combination and it might be objectively existed that the interaction effects between the four parameters: stimulating site, frequency, intensity and pulse number. Furthermore, we predicted that the mechanism of rTMS might be related with the neuroplasticity in three brain networks: default mode network (DMN), dorsal attention network (DAN), central executive network (CEN). Hence, the prospective, single-center, orthogonally-designed, triple-blind randomized controlled trial will be conducted and will be aimed to explore the rTMS optimum parameters regimen for PSCI based on stimulating site, frequency, intensity and pulse number and the neuroimaging mechanism in default mode network (DMN), dorsal attention network (DAN), central executive network, (CEN) following rTMS treatment in PSCI patients. All participants will be assigned averagely to one of nine paralleled active rTMS groups which are designed according to the $L_9(3^4)$ orthogonal array with four different parameters: stimulating site, frequency, intensity and pulse number. This trial protocol will be written in adherence to the standard protocol items: recommendations for interventional trials (SPIRIT) 2013 guidelines [14].

## Methods

### 1. Patients

**(1) Study setting.** This study will be carried out at the West China Hospital of Sichuan University (Chengdu, China). The post-stroke cognitive impairment patients in the wards of the departments of neurology and rehabilitation medicine from December 2021 to December 2023 will be screened for eligibility and finally 45 patients will be enrolled in this trial.

**(2) Eligibility criteria.** The study researchers will execute the eligibility screen in accordance with the inclusion and exclusion criteria. All the participants should sign an informed consent after eligibility confirmation.

Inclusion criteria:

- Male or female patients between 18 and 75 years of age;

- Right-handed;

- Patients who meet the diagnostic criteria for vascular cognitive impairment reported in the Guidelines from the Vascular Impairment of Cognition Classification Consensus Study [15] and the Chinese vascular cognitive impairment guideline 2019 [16] as which mainly be concluded as: (1) Cognitive impairment at least one domain in attention and executive, memory, language and visuospatial function confirmed by clinical and neuropsychological evaluation (MoCA ≤ 24) with or without the impairment in instrumental activities of daily life (IADL); (2) Imaging evidence due to stroke confirmed by magnetic resonance imaging (MRI); (3) Cognitive impairment should be independent of the motor/sensory sequelae of the vascular event and with a clear temporal relationship (within 6 months and lasted for at least 3 months) between a vascular event and onset of cognitive deficits;

- No severe aphasia (Unable to accomplish cognitive tests);

- Normal cognitive functions before stroke;

- Voluntary participation and signed the informed consent (Signed by the patient or the authorized person).

Exclusion criteria:

- Cognitive impairment due to primary or secondary neurological disease, such as normal cranial pressure hydrocephalus, frontotemporal dementia, Parkinson's disease, multiple sclerosis, encephalitis and delirium;

- Cognitive impairment due to depression, schizophrenia, bipolar disorder, psychotic disorder, vitamin D deficiency, toxicosis or else systematic diseases;

- Drug/alcohol abuse/dependence within the last 3 months of first recognition of impairment.

- rTMS treatment contraindications such as epilepsy patients, pregnant or lactating women or with a metal or electric implanted device (eg. deep brain stimulator, ventriculoperitoneal shunt, aneurysm clip, pacemaker, cochlear, surgical staples on the scalp).

- MRI contraindications (such as metal implants or claustrophobia)

- participating in a concurrent pharmacological or nonpharmacologic treatment research.

**(3) Participant timeline.** Study will commence with the eligibility screen for participants. Once eligibility is confirmed, the patient will sign an informed consent and be allocated to one

| | STUDY PERIOD | | | | | | | |
|---|---|---|---|---|---|---|---|---|
| | Enrolment | Allocation | Post-allocation | | | | Close-out | Follow-up |
| **TIMEPOINT** | **-$w_1$** | **0** | **$w_1$** | **$w_2$** | **$w_3$** | **$w_4$** | **$w_5$** | **$w_{17}$** |
| **ENROLMENT:** | | | | | | | | |
| **Eligibility screen** | X | | | | | | | |
| **Informed consent** | X | | | | | | | |
| **Allocation** | | X | | | | | | |
| **INTERVENTIONS:** | | | | | | | | |
| *[rTMS 1]*: TPC; 5 Hz; 100%RMT; 1500 pulses | | | X←———————→X | | | | | |
| *[rTMS 2]*: IFG; 20 Hz; 90%RMT; 1500 pulses | | | X←———————→X | | | | | |
| *[rTMS 3]*: IFG; 10 Hz; 100%RMT; 1000 pulses | | | X←———————→X | | | | | |
| *[rTMS 4]*: TPC; 10 Hz; 90%RMT; 2000 pulses | | | X←———————→X | | | | | |
| *[rTMS 5]*: DLPFC; 5 Hz; 90%RMT; 1000 pulses | | | X←———————→X | | | | | |
| *[rTMS 6]*: IFG; 5 Hz; 110%RMT; 2000 pulses | | | X←———————→X | | | | | |
| *[rTMS 7]*: DLPFC; 10Hz; 110%RMT; 1500 pulses | | | X←———————→X | | | | | |
| *[rTMS 8]*: TPC; 20 Hz; 110%RMT; 1000 pulses | | | X←———————→X | | | | | |
| *[rTMS 9]*: DLPFC; 20Hz; 100%RMT; 2000 pulses | | | X←———————→X | | | | | |
| *[RMC]*: Individual cognitive training | | | X←———————→X | | | | | |
| **ASSESSMENTS:** | | | | | | | | |
| **Demographics and clinical characteristics** | | X | | | | | | |
| **MoCA; TMT; MBI; LIADL; SS-QOL** | | X | | | | | X | X |
| **fMRI Scan** | | X | | | | | X | |

*NOTES:* DLPFC: Dorsolateral prefrontal cortex; fMRI: Functional magnetic resonance imaging; IFG: Inferior frontal gyrus; LIADL: Lawton instrumental activities of daily living; MBI: Modified Barthel index; MoCA: Montreal cognitive assessment; RMC: Routine medical care; RMT: Resting motor threshold; rTMS: Repetitive transcranial magnetic stimulation; SS-QOL: Stroke-specific quality of life; TMT: Trail marking test; TPC: Temporoparietal cortex

**Fig 1. The schedule of the trial.**

of nine active rTMS group. Clinical assessments and functional magnetic resonance imaging (fMRI) scan will be performed at baseline (W0), termination of 20 rTMS sessions (W5) and the follow-up (W17). In the any phase of the study procedure, we will record the number and reasons for any participants who will be excluded or any who will decline consent or withdraw from the study. The SPIRIT schedule and the flowchart of the study are presented in Fig 1.

**(4) Sample size.** Because no previous orthogonally designed study provides a significant difference of MoCA or MMSE scores for rTMS treatment to PSCI, which would allow us to accurately estimate the adequate sample size for this trial, we decided to recruit 45 PSCI patients who will be randomly divided into nine groups and three subgroups based on orthogonal design (5 patients per group, 15 patients per stimulating site subgroup), in consideration of the research grant, the recruitment feasibility and the recommended sample size in neuroimaging research (medians:14.5) [17].

**(5) Recruitment.** This trial aims to recruit 45 in- or outpatients with PSCI in the West China Hospital of Sichuan University who are recommended by the rehabilitation physicians, therapists, or any other medical practitioners and willing to receive rTMS treatment. Once the oral consent to take part in the study is confirmed, written and verbal information about the study aim and procedures is provided to all the volunteer participants. There are no biological specimens collecting for storage and anticipated harm for the participants throughout the study. All the personal information and any other data about the potential or enrolled patients will be traceable in the medical records and available from the corresponding author on request in order to protect confidentiality. After the written informed consents are obtained from the eligibility-screened participants, the study will commence with the baseline assessment and then the randomized allocation.

**Table 1. The orthogonal experimental factors and levels of rTMS.**

| Level | Experimental factor | | | |
|---|---|---|---|---|
| | A (Site) | B (Frequency, Hz) | C (Intensity, %RMT) | D (Pulse) |
| 1 | DLPFC | 5 | 90 | 1000 |
| 2 | IFG | 10 | 100 | 1500 |
| 3 | TPC | 20 | 110 | 2000 |

Note: DLPFC: Dorsolateral prefrontal cortex; IFG: Inferior frontal gyrus; RMT: Resting motor threshold; rTMS: Repetitive transcranial magnetic stimulation; TPC: Temporoparietal cortex

**(6) Randomization and blinding.** The random numbers sequence is generated using SPSS software in computer and all eligible patients were equal assigned to one of nine active rTMS treatment group based on four rTMS parameters. The random allocation and detailed rTMS protocol are only known by the rTMS physiotherapists and blinded to physicians, patients and other researchers (such as outcome assessors, data analysts). The rTMS physiotherapists are not involved in any other study work such as patient recruitment, randomization, allocation, outcome assessment and data analysis.

## 2. Interventions

**(1) rTMS protocol.** The rTMS treatments were delivered by a magnetic simulator (YRD CCY-I, YIRUIDE medical equipment co., LTD, Wuhan, China) with a figure-8 coil. The rTMS protocol was designed as nine combinations of four parameters at three different levels in each parameter: stimulating site (the dorsolateral prefrontal cortex, DLPFC, Brodmann areas 9 and 46; the inferior frontal gyrus, IFG, Brodmann areas 44 and 45; the temporo-parietal cortex, TPC, Brodmann areas 39 and 40), frequency (5Hz, 10Hz, 20Hz), intensity (90%, 100%, 110% of the resting motion threshold, RMT) and pulse number (1000, 1500, 2000), the detailed information are showed in Tables 1 and 2. The aforementioned rTMS parameters and different levels were predetermined based on the previous studies involving in rTMS treatment to cognition impairment duo to various neurological and psychological diseases. Once the PSCI patients were recruited, they were randomly allocated into one of nine active rTMS groups, a total of 20 sessions (four weeks, five consecutive daily sessions and

**Table 2. The orthogonal array L9(34).**

| Groups | Site (A) | Frequency (B) | Intensity (C) | Pulse (D) | rTMS protocols | Sample Size |
|---|---|---|---|---|---|---|
| 1 | TPC | 5 Hz | 100%RMT | 1500 | $A_3B_1C_2D_2$ | 5 |
| 2 | IFG | 20 Hz | 100%RMT | 1000 | $A_2B_3C_2D_1$ | 5 |
| 3 | IFG | 10 Hz | 90%RMT | 1500 | $A_2B_2C_1D_2$ | 5 |
| 4 | TPC | 10 Hz | 110%RMT | 1000 | $A_3B_2C_3D_1$ | 5 |
| 5 | DLPFC | 5 Hz | 90%RMT | 1000 | $A_1B_1C_1D_1$ | 5 |
| 6 | IFG | 5 Hz | 110%RMT | 2000 | $A_2B_1C_3D_3$ | 5 |
| 7 | DLPFC | 10 Hz | 100%RMT | 2000 | $A_1B_2C_2D_3$ | 5 |
| 8 | TPC | 20 Hz | 90%RMT | 2000 | $A_3B_3C_1D_3$ | 5 |
| 9 | DLPFC | 20 Hz | 110%RMT | 1500 | $A_1B_3C_3D_2$ | 5 |

Note: $A_1$: DLPFC; $A_2$: IFG; $A_3$: TPC; $B_1$: 5Hz; $B_2$: 10Hz; $B_3$: 20Hz; $C_1$: 90%RMT; $C_2$: 100%RMT; $C_3$: 110%RMT; $D_1$: 1000; $D_2$: 1500; $D_3$: 2000; DLPFC: Dorsolateral prefrontal cortex; IFG: Inferior frontal gyrus; RMT: Resting motor threshold; rTMS: Repetitive transcranial magnetic stimulation; TPC: Temporoparietal cortex

two days of rest per week) active rTMS treatments with different parameters combination will be administered by two therapists with more than 5 work experience at the neuromodulating center of the rehabilitation medicine department, West China Hospital, Sichuan University. In order to improve the patient's adherence to the intervention protocols, once every rTMS session was completed, a predesigned treatment record card should be filled and signed by the rTMS therapists and the patients or their authorized persons and ultimately returned to the study researchers.

The detailed information of parameters and grouping were as following:

Group 1: TPC; 5 Hz; 100%RMT; 1500 pulses (10s × 20 trains, 10s interval)

Group 2: IFG; 20 Hz; 90%RMT; 1500 pulses (3s × 25 trains, 20s interval)

Group 3: IFG; 10 Hz; 100%RMT; 1000 pulses (4s × 25 trains, 20s interval)

Group 4: TPC; 10 Hz; 90%RMT; 2000 pulses (8s × 25 trains, 20s interval)

Group 5: DLPFC; 5 Hz; 90%RMT; 1000 pulses (8s × 25 trains, 20s interval)

Group 6: IFG; 5 Hz; 110%RMT; 2000 pulses (20s × 20 trains, 20s interval)

Group 7: DLPFC; 10Hz; 110%RMT; 1500 pulses (6s × 25 trains, 20s interval)

Group 8: TPC; 20 Hz; 110%RMT; 1000 pulses (2s × 25trains, 20s interval)

Group 9: DLPFC; 20Hz; 100%RMT; 2000 pulses (5s × 20 trains, 20s interval)

**(2) Neuro-navigation.** The rTMS stimulating sites were identified on the 3D brain reconstruction using a neuro-navigation system (Brainsight, Rogue Research Inc., Montreal, QC, Canada). Before the experiment, T1 weighted images were individually obtained from each patients using rapid acquisition gradient echo sequence (Philips Ingenia 3 Tesla, MPRAGE, TR/TE = 2000ms/4.6 ms, voxel size = 0.98 mm × 0.98 mm × 1.2 mm, the field of view = 250 mm × 250 mm × 240 mm, 200 sagittal slices). To localize the rTMS sites, the method described by Mylius et al. [18] were used. Briefly, the boundary of the middle frontal gyrus (MFG) was firstly drawn on the neuro-navigated 3D brain reconstruction, then the DLPFC target was determined on the middle of the bonder line between the anterior one third and the posterior two thirds of the MFG. Similarly, the IFG target was determined on the middle of the transverse intermediate line of the inferior frontal gyrus (IFG), the TPC target was determined on the middle of the border line between the supramarginal gyrus (SG) and the angular gyrus (AG). All the rTMS stimulating sites were selected on the affected cerebral hemisphere.

**(3) RMT.** The resting motor threshold (RMT) is defined as the lowest stimulation intensity to induce a, at least 50% of the time in a finite number of trials (typically 10 trials).

The RMT was determined as the minimum stimulating intensity necessary to elicit an overt motor response (motor evoked potential, MEP, larger than 50 µV) in the right abductor pollicis brevis (APB) at least 5 of 10 times [19].

**(4) Route medical cares.** The route medical cares based on the individualized illness for each patient are permitted but the medications for improving cognition such as Pfizer's Aricept were prohibited during the trial.

**(5) Discontinuing criteria.**

• Patients who simply wishes to stop participation;

• Patients who could not undergo the baseline assessment;

- Patients who did not complete the rTMS treatment sessions;

- Patients who suffered from worsening symptoms.

## 3. Outcomes

The primary outcomes are the Montreal Cognitive Assessment Beijing Version (MoCA-BJ) and the Trail-Making Test (TMT). The secondary outcomes are the modified Barthel index (MBI), the Lawton Instrumental Activities of Daily Living Scale (LIADL), the stroke-specific quality of life (SS-QOL), the functional magnetic resonance imaging (fMRI) scan and the adverse events. All measurements of clinical assessments were administered three times: pre-treatment (baseline), post-treatment, and at a three-month follow-up. All of the cognitive assessments were performed by a trained neuropsychologist who was blind to the treatment status of the participants throughout the study. The MRI scan were conducted at baseline and at the end of rTMS treatment.

**(1) MoCA-BJ.** The MoCA- BJ scale is one of the Chinese versions of the Montreal cognitive assessment used to assess the general cognitive function [20], which has high sensitivity and specificity for screening cognitive impairment from patients with stroke [21].

**(2) TMT.** The trail marking test (TMT) was firstly used in 1944 for America army individual intelligence assessment [22]. Nowadays, it is widely employed as a diagnostic tool for visual attention and executive functioning and the processing speed for neuropsychological diseases [23, 24]. It consists of two parts in which the subject is instructed to connect a set of 25 circles containing numbers (Part A) or numbers and letters (Part B) arrayed pseudorandomly on a letter-size sheet of paper as fast as possible while still maintaining accuracy.

**(3) ADL.** The basic activities of daily living are assessed with the modified Barthel Index (MBI) [25], which is used to assess the individual self-care performance and derived from the Barthel index [26]. In this study, we used the Chinese version of MBI [27], it also consists of ten items: personal hygiene, bathing, feeding, toileting, stair climbing, dressing, bowel control, bladder control, ambulation or wheelchair and chair-bed transfer with total 100 scores indicating completed independence. The instrumental activities of daily living are evaluated with the Lawton Instrumental Activities of Daily Living Scale (LIADL) [28], which is consists of eight items: ability to use telephone, shopping, food preparation, housekeeping, laundry, mode of transporation, responsibility for own medications, ability to handle finances with higher scores representing more independent.

**(4) SS-QOL.** The Stroke-Specific Quality of Life Scale (SS-QOL), the first health-related QOL measure for stroke patients, was made available in 1999 [29]. It consists of 12 domains, 78 items with total 390 scores indicating a quality of life and is commonly used to evaluate the treatment methods, the rehabilitation programs and the burden of disease in stroke related studies [30].

**(5) MRI scan and data preprocessing.** MRI scanned on a 3.0 Tesla MRI system (GE Medical Systems, Waukesha, WI, USA) at the West China Hospital of Sichuan University. The structural MRI (High-resolution 3-demention T1-weighted, gradient echo, GRE) sequence (TR/TE = 8.5 ms/3.2 ms; flip angle = 12˚; field of view = $256 \times 256$ mm$^2$; matrix = $256 \times 256$; slice thickness = 1 mm; and 176 axial slices, scanning time = 4min33s) images were obtained with the same parameters in neuro-navigation. The resting-state functional MRI images were conducted with T2*- weighted Gradient echo—echo planar imaging (GRE-EPI) sequence (TR/TE = 2000 ms/35 ms; flip angle = 90˚; field of view = $230 \times 230$ mm$^2$; matrix = $64 \times 64$; slice thickness = 3.6 mm; and 35 axial slices, dummy samples = 5; statistical samples = 235, scanning time = 8 min) at pre- and post- treatment. When the fMRI

scanning, the patients were required to close their eyes and keep awake and relax. All of the rs-fMRI data were preprocessed using Statistical Parametric Mapping (SPM8, http://www.fil.ion.ucl.ac.uk/spm) and Data Processing Assistant for Resting-State fMRI (DPARSF) [31]. Briefly, the volumes in the first 5 s were discarded for signal equilibrium. The remaining data were corrected for head motion. These data were then spatially normalized to the Montreal Neurological Institute (MNI) space and resampled to 3-mm isotropic voxels. We perform spatial smoothing with an $8 \times 8 \times 8$ mm$^3$ Gaussian kernel only for the ALFF analysis. Next, the linear trend of the data was removed, and temporal band-pass filtering (0.01–0.1 Hz, only for the functional connectivity (FC) analysis) was performed to reduce the effects of low-frequency drift and high-frequency physiological noise. Finally, six head motion parameters and three potential nuisance signals, including the cerebrospinal fluid, white matter and global signals, were removed from the time course of each voxel using multiple linear regression.

(6) **ALFF and FC.** The amplitude of low frequency fluctuation (ALFF) and functional connectivity (FC) will be used to explore different brain effects in the three stimulating -site subgroups.

The ALFF refers to the low frequency fluctuation of BOLD signal of the brain voxel, which will be calculated with DPARSF package after data preprocessing. The generated ALFF maps will be used for statistical analysis.

FC will be used to explore the functional connectivity changes in three brain networks (default mode network, DMN; dorsal attention network, DAN; central executive network, CEN) after treatment, which were obtained by meta-analysis (http://www.neurosynth.org/), were used as imaging indicators to analyze according to the methods described by Tomasi et al. [32]. First, Independent component analysis (ICA) will be used to extract DMN, DAN and CEN networks of the three groups, and then the brain regions with differences in the corresponding network will be found by visual observation, which will be used as regions of interest (ROI) to conduct Pearson correlation with the whole brain voxel. The FC maps obtained will be used for statistical analysis.

(7) **Adverse events.** Adverse Events (AE) during or $\leq$1h after session, including headache, scalp dysesthesia/paresthesia at stimulation site, muscle pain of temporal or neck muscles and seizures, will be documented after each treatment session and during the whole treatment period.

## 4. Statistical analyses

The data of MoCA-BJ, TMT, MBI, LIADL and SS-QOL were shown as mean ± standard deviation (SD) and analyzed by the range analysis and one-way analysis of variance (ANOVA), which was used the SPSS 22.0 software (IBM, Armonk, NY, USA, 2020). The range analysis was used to determine the influence of each parameter at a specific level on the effect indicator and obtain the optimum combination of rTMS parameters. The one-way ANOVA was used to determine the statistical significance in nine different parameters' combination of rTMS. P < 0.05 was considered significant.

For ALFF and FC maps, paired T-test will be used for intra-group comparison and one-way ANOVA analysis will be used for inter-group comparison in the three stimulating site subgroups. Then multiple comparison correction will be performed using Gaussian random field theory (GRF) with voxel $p < 0.005$, cluster $p < 0.05$. The Pearson correlation analyses were used to calculate the correlations between the changes in ALFF and FC with the clinical symptom scores at baseline and the end of the rTMS treatment ($p < 0.05$).

## 5. Ethics approval and declaration of interests

The trial will be conducted in accordance with the ethical principles outlined in the Declaration of Helsinki, 1996 and was approved by the Ethics Committee on Biomedical Research, West China Hospital of Sichuan University (2020–1121).

## 6. Trial status

This trial was registered on the website of Chinese Clinical Trial Registry (Registration number: ChiCTR2100049625; https://www.chictr.org.cn/). At the writing of this paper, we are recruiting subjects. The study began in December 2021 and planned to complete in December 2023.

## Discussion

As a non-invasive brain stimulation technique, the rTMS uses high intensity magnetic field pulses to modulate neural activity and in turn modifying the brain function. Over the past few decades, rTMS is widely applied into the cognition impairment due to many neurological, psychiatrical and psychological diseases, such as stroke, Alzheimer's disease, Parkinson's disease, schizophrenia, obsessive compulsive disorder. Several meta-analyses and RCT have showed that the safe and beneficial value of rTMS in cognitive function rehabilitation [10, 33–35]. Given the more and more application of this method in clinical and research settings, it is crucial to understand what is the optimal parameters of rTMS for cognition impairment and how the rTMS modulates brain function via the high intensity magnetic field pulses. Fortunately, some researchers have realized these two key points. Beynel L et al. thought that the rTMS parameters include a vast composition of spatial and temporal parameters: coil geometry, stimulation target, stimulation intensity; pulse waveform, pulse train frequency, number of pulses, etc [36]. while it does not be summed precisely from the perspective of clinical application. Based on the previous literature related to multiple sessions of rTMS for cognition impairment, we hold the opinion that the most important four parameters of rTMS are stimulating site, stimulating frequency, stimulating intensity and pulse number. Orthogonal design is a powerful method for comparative effectiveness research, especially for intervention with multiple parameters, it could significantly reduce the experiment numbers and alleviate the research workloads [37]. In rehabilitation medicine, many physical modalities are delivered with a specific equipment by presetting several parameters in the meantime. It is important that these parameters should be investigated at the same time when we are going to assess the efficacy and safety of this intervention. For example, the rTMS will be delivered to a patient with cognition impairment, the stimulating site, frequency, intensity and pulses should be predetermined meanwhile. However, in the previous studies, it was common that only one parameter, such as frequency, be compared with different level, or be compared between active and sham rTMS. It is not entirely and exactly consistent with clinical practice. Hence, in order to investigate the optimum parameters of clinical value of rTMS for post-stroke cognition impairment, the present study is designed orthogonally with $L_9(3^4)$ orthogonal array based four parameters with three different level in each parameter (sites: DLPFC, IFG, TPC; frequency: 5Hz, 10Hz, 20Hz; intensity: 90%RMT, 100%RMT, 110%RMT; pulse number: 1000 pulses, 1500 pulses, 2000 pulses). On the other hand, although many studies showed that the modulatory effect of rTMS on brain function, ig. enhancing the neural activity and the connectivity between different brain regions [38], is correlated to the improvement of cognition function [39], the exact neurological mechanism has not been understood very well. Giacomo Koch et al. showed that the rTMS stimulation on precuneus enhances memory and neural activity and connectivity in default mode network [40]. Recep A Ozdemir et al. Showed that

the default mode network (DMN) and the dorsal attention network brain network (DMN/ DAN) specificity of rTMS activations is correlated with cognitive performance in healthy volunteers [41]. Reza Kazemi et al. showed that the bilateral rTMS stimulation on DLPFC changes resting state in CEN, DMN brain networks and cognitive function in patients with bipolar depression [42]. However, whether the brain networks related to attention and executive function (CEN, DMN, DAN) is enhanced activity and connectivity and is correlated with the cognitive improvement following the rTMS stimulation in patients with post-stroke cognition impairment are still unclear. Therefor, in the current study, the amplitude of low-frequency fluctuation (ALFF) and the functional connectivity (FC) of the resting-state fMRI signal has been adopted to reflect the regional brain network (CEN, DMN, DAN) activity and connectivity and further our understanding of the complex cognitive function recovery process underlying metaplasticity with rTMS, demonstrate how the adaptive metaplasticity can contribute to the cognition impairment, and show that whether the maladaptive metaplasticity play a role in the cognitive pathophysiology [43].

Although the present study brings forth a new orthogonal design to probe the optimal rTMS parameters treating PSCI, it still has many limitations, including the small sample size and the single-center study, due to the recruitment difficulty in the background of coronavirus disease 2019 (COVID-19) management. Besides, the relatively simple parameter level design and the heterogeneity of the oral medications in PSCI patients might be also the pitfalls in the measures of cortical excitability and their response to rTMS treatments [44]. A multi-center, large-sample and more rigorously designed study may be a possible solution in future.

## Supporting information

**S1 File. SPIRIT-Checklist.**
(DOC)

**S2 File. Ethical approval document-Chinese.**
(PDF)

**S3 File. Ethical approval document-English.**
(DOCX)

**S4 File. Interventional clinical study program-Chinese.**
(DOC)

**S5 File. Interventional clinical study program-English.**
(DOC)

**S6 File. Funding letter-Chinese.**
(PDF)

**S7 File. Funding letter-English.**
(DOCX)

## Author Contributions

**Data curation:** Shuai He.

**Funding acquisition:** Ling-Xin Li.

**Investigation:** Jing-Kang Lu, Bao-Jin Li, Qian Wen.

**Methodology:** Qiang Gao.

**Software:** You-Jin Zhao.

**Supervision:** Cheng-Qi He, Shi-Hong Zhang.

**Writing – original draft:** Ling-Xin Li.

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
