## [Decision Letter · Decision Letter 0]

4 May 2022

PONE-D-21-30462The optimum parameters and neuroimaging alteration in DMN, DAN, CEN of rTMS to PSCI patients, a protocol of an orthogonally-designed triple-blind RCTPLOS ONE

Dear Dr. LI,

Thank you for submitting your manuscript to PLOS ONE. After careful consideration, we feel that it has merit but does not fully meet PLOS ONE’s publication criteria as it currently stands. Therefore, we invite you to submit a revised version of the manuscript that addresses the points raised during the review process.

Please see the comments from two reviewers below, who have both suggested revisions to enhance the clarity and presentation of the study. Please ensure that you provide thorough answers to their requests, and that you copyedit the manuscript before resubmission. Please pay attention to the request to further explain the consent procedures.

We look forward to receiving your revised manuscript.

Kind regards,

Hanna Landenmark

Staff Editor

PLOS ONE

Journal Requirements:

This work was supported by the Post-Doctor Research Project, West China Hospital, Sichuan University (No. 2020HXBH018); the Chengdu Science and Technology Bureau (No. 2019-YF05-00061-SN); and the Health Department of Sichuan Province (No. S15063). The funders played no role in the design, conduct, or reporting of this study.

The Post-Doctor Research Project, West China Hospital, Sichuan University (2020HXBH018); the Chengdu Science and Technology Bureau (No. 2019-YF05-00061-SN); and the Health Department of Sichuan Province (No. S15063) funded this research.

The funders had and will not have a role in study design, data collection and analysis, decision to publish, or preparation of the manuscript.

5. Please include your tables as part of your main manuscript and remove the individual files. Please note that supplementary tables (should remain/ be uploaded) as separate "supporting information" files"

Reviewers' comments:

Reviewer's Responses to Questions

**Comments to the Author**

1. Does the manuscript provide a valid rationale for the proposed study, with clearly identified and justified research questions?

Reviewer #1: Yes

Reviewer #2: Yes

2. Is the protocol technically sound and planned in a manner that will lead to a meaningful outcome and allow testing the stated hypotheses?

Reviewer #1: Yes

Reviewer #2: Partly

3. Is the methodology feasible and described in sufficient detail to allow the work to be replicable?

Reviewer #1: Yes

Reviewer #2: No

4. Have the authors described where all data underlying the findings will be made available when the study is complete?

Reviewer #1: Yes

Reviewer #2: Yes

5. Is the manuscript presented in an intelligible fashion and written in standard English?

Reviewer #1: No

Reviewer #2: No

6. Review Comments to the Author

You may also provide optional suggestions and comments to authors that they might find helpful in planning their study.

Reviewer #1: The authors deal with a relevant and timely topic, i.e., the optimum parameters and neuroimaging alteration in default mode network (DMN), dorsal attention network (DAN), and central executive network, (CEN) of repetitive transcranial magnetic stimulation (rTMS) in patients with post-stroke cognitive impairment (PSCI). To this end, they proposed a single-center, orthogonally-designed, triple-blind randomized controlled trial in 45 patients, who will be randomly assigned to one of 9 active rTMS groups based on four stimulation paraments: stimulating site, frequency, intensity, and pulse number. Neuropsychological, activities of daily living, quality of life, and functional neuroimaging evaluations will be also performed at baseline, post-rTMS, and 3 months after rTMS. The authors concluded that this study will evaluate the optimum parameters of rTMS for PSCI patients with and will explore the alteration of neural function in DMN, DAN, and CEN brain network. This would facilitate the standardized application of rTMS in cognition impairment rehabilitation.

Overall, the protocol is nicely conceived and designed; the methods are sufficiently described and might disclose useful translational data. I have some comments, especially in the Introduction and in the Discussion. Moreover, language should be edited by a native-English speaker.

Title: please reduce or spell out the use of abbreviation.

Abstract: please include a brief rationale of the protocol; the abbreviation PSCI should be spelled out.

Introduction: please provide a short overview on the pathophysiology and neurobiology of PSCI and, more in general, of vascular cognitive impairment (for a recent comprehensive review, see PMID: 32340195). In particular, the exact relationships between vascular lesion, cognition, and neuroplasticity are not completely understood, although converging findings point out the possibility to identify a panel of markers able to predict cognitive deterioration in the so-called "brain at risk" for vascular or mixed dementia. This will be of pivotal importance when designing trials of disease-modifying drugs or non-pharmacological approaches, such as non-invasive neuromodulatory techniques (including rTMS) in dementia, as very recently reviewed (PMID: 34482205).

Introduction: please state the experimental hypothesis: what did the authors expect? Why and how?

Methods: did the authors include in the protocol also a sham (fictitious) stimulation procedure?

Methods: although a total of 45 patients is acceptable, 5 subjects per group is a rather small sample size.

Discussion: in the context of the proposed protocol, it should be mentioned that, in some patients plasticity may be considered as an adaptive response to disease progression (thus allowing the preservation of motor programming and execution) but a maladaptive response in others. Please also mention the involvement of metaplasticity in neurological and neuropsychiatric disorders, including stroke and PSCI (PMID: 34276553).

Discussion: pitfalls and limitations of both protocol and technique should be included, together with possible solutions and future research agenda (for some suggestions, please see PMID: 33193753).

Reviewer #2: Dear Editor,

Thank you for the opportunity to provide a review of Manuscript PONE-D-21-30462 entitled "The optimum parameters and neuroimaging alteration in DMN, DAN, CEN of rTMS to PSCI patients, a protocol of an orthogonally-designed triple-blind RCT". My comments relate primarily to the adequacy of the implementation and reporting of epidemiologic and statistical procedures.

The quality of the technical English is inconsistent. There are grammatical errors scattered throughout the text. The authors need to perform a thorough round of copyediting to correct these. Nevertheless, these did not affect my evaluation of the manuscript.

# Major Issues

Page 10: "active rTMS groups in 1:1 ratio". This ratio is wrong since there are nine treatment groups, not two.

Page 10: "This trial will be conducted in line with the consolidated standards of reporting trials (CONSORT) statement...". CONSORT should NOT be used to guide the conduct of trials. These are REPORTING standards. CONSORT does not tell you how to conduct a study. They tell you how to describe it in a consistent fashion. Please change this statement.

Page 11: "No severe aphasia". How will aphasia be assessed and what is the criteria for severe aphasia used here?

Page 12: "informed consent". In this study where patients are cognitively impaired, how will the capacity to understand the study and to give proper informed consent be assessed?

Page 12: The description of the sample size calculations is inappropriate and cannot be confirmed because there is no replicable information provided by the authors. This must be corrected.

Page 19: The use of one-way ANOVA to analyse paired data (MRI ALFF and ACD) is highly inappropriate. This must be changed.

Page 19: The level of significance must be stated.

*IMPORTANT* There is no assessment of safety. This is a major oversight and needs to be added. This is non-negotiable.

*IMPORTANT* Why is there no sham group?

# Recommendation

I am unable to support the approval of this manuscript for publication in the journal until these issues are considered.

Thank you.

7. PLOS authors have the option to publish the peer review history of their article (what does this mean?). If published, this will include your full peer review and any attached files.

Reviewer #1: No

Reviewer #2: No

---

## [Author Response · Author response to Decision Letter 0]

30 May 2022

Response to Reviewers PONE-D-21-30462

Dear reviewers:

We have revised the manuscript and replied the 17 items of comments (2 reviewers) point-to-point as below. The detailed change information is showed in the “Main text-Revised Manuscript with Track Changes”.

Thank you for your timely reply!

Your sincerely

Lingxin Li

West China Hospital, Chengdu, China

2022-05-22

Reviewer #1: 

Item 1:

Title: please reduce or spell out the use of abbreviation.

Reply:

We have changed and spelled out all of the abbreviations in title. (Line 2 - 12)

Item 2:

Abstract: please include a brief rationale of the protocol; the abbreviation PSCI should be spelled out.

Reply:

We explained the reason and rationale of the protocol and the changes were showed in the objective part of Abstract. The abbreviation PSCI is also be spelled out. (Line 69 – 73; Line 76)

Item 3:

Introduction: please provide a short overview on the pathophysiology and neurobiology of PSCI and, more in general, of vascular cognitive impairment (for a recent comprehensive review, see PMID: 32340195). In particular, the exact relationships between vascular lesion, cognition, and neuroplasticity are not completely understood, although converging findings point out the possibility to identify a panel of markers able to predict cognitive deterioration in the so-called "brain at risk" for vascular or mixed dementia. This will be of pivotal importance when designing trials of disease-modifying drugs or non-pharmacological approaches, such as non-invasive neuromodulatory techniques (including rTMS) in dementia, as very recently reviewed (PMID: 34482205).

Reply:

We added a short overview of the pathophysiology and neurobiology of PSCI in the introduction part and added three references [5-7]. (Line 107 – 115)

Item 4:

Introduction: please state the experimental hypothesis: what did the authors expect? Why and how?

Reply:

We added the hypothesis of the study in the introduction part. (Line 126 – 132)

Item 5:

Methods: did the authors include in the protocol also a sham (fictitious) stimulation procedure?

Reply:

The objective of this study is to probe what the optimal combination of four parameters is, rather than whether the rTMS is valid for the recovery of cognitive function in PSCI patients, so that the sham stimulation control is not designed.

Item 6:

Methods: although a total of 45 patients is acceptable, 5 subjects per group is a rather small sample size.

Reply:

Under the background of the epidemic COVID-19 in the world, the researchers consider that the subject recruitment is difficult. Secondly, although 5 subjects per group is a rather small sample size, the test efficacy is fit with the range and the one-way ANOVA analysis for an orthogonally-designed study. Thirdly, the fMRI data will be analyzed based on three stimulating site subgroups with 15 subjects per group and this met to the recommended sample size medians (14.5 per group) in a literature [Denes Szucs, John Pa Ioannidis. Sample size evolution in neuroimaging research: An evaluation of highly-cited studies (1990-2012) and of latest practices (2017-2018) in high-impact journals. Neuroimage. 2020; 221:117164. PMID: 32679253, [18]]. Hence, the research team decided the sample size as 45 patients in the study. (Line 207 – 214)

Item 7:

Discussion: in the context of the proposed protocol, it should be mentioned that, in some patients plasticity may be considered as an adaptive response to disease progression (thus allowing the preservation of motor programming and execution) but a maladaptive response in others. Please also mention the involvement of metaplasticity in neurological and neuropsychiatric disorders, including stroke and PSCI (PMID: 34276553).

Reply:

It is true that the maladaptive neuroplasticity which might contribute to the dysfunctional remodeling of some specific neural networks in neurological or neuropsychiatric diseases.

We discussed it in the discussion part in our manuscript and added a reference [45]. (Line 498 – 503)

Item 8:

Discussion: pitfalls and limitations of both protocol and technique should be included, together with possible solutions and future research agenda (for some suggestions, please see PMID: 33193753).

Reply:

We evaluated discreetly our study protocol again and discussed the pitfalls and limitations at the end of the discussion part, including the small sample size, the single center study etc. in the end of the Discussion part and added a reference [46]. (Line 504 – 511)

 Reviewer #2: 

# Major Issues 

Item 9:

Page 10: "active rTMS groups in 1:1 ratio". This ratio is wrong since there are nine treatment groups, not two. 

Reply:

This mistake has been corrected as “All participants will be assigned averagely to one of nine paralleled active rTMS groups…” (Line 140 – 141)

Item 10:

 Page 10: "This trial will be conducted in line with the consolidated standards of reporting trials (CONSORT) statement...". CONSORT should NOT be used to guide the conduct of trials. These are REPORTING standards. CONSORT does not tell you how to conduct a study. They tell you how to describe it in a consistent fashion. Please change this statement. 

Reply:

It has been revised as “This trial protocol will be written in adherence to the standard protocol items: recommendations for interventional trials (SPIRIT) 2013 guidelines [15].”

The reference [12] has also been removed. (Line 143 – 147)

Item 11:

Page 11: "No severe aphasia". How will aphasia be assessed and what is the criteria for severe aphasia used here? 

Reply:

No optimal aphasia criteria could evaluate which extent of aphasia is severe aphasia in this trial conducted in China. In practice, we defined the severe aphasia as the global aphasia or in the extent of which will hinder the subjects to complete the MMSE or MoCA assessment pre- or post- treatment. (Line 175 – 176)

Item 12:

Page 12: "informed consent". In this study where patients are cognitively impaired, how will the capacity to understand the study and to give proper informed consent be assessed? 

Reply:

The informed consent will be signed by the patient or the authorized person. (Line 178 – 179)

Item 13:

Page 12: The description of the sample size calculations is inappropriate and cannot be confirmed because there is no replicable information provided by the authors. This must be corrected. 

Reply:

Because no previous orthogonally designed study provided a significant difference of MoCA or MMSE scores for rTMS treatment to PSCI to estimate the sample size. Hence, we decided to recruit 45 PSCI patients in consideration of the research grant, the recruitment feasibility and the statistic power of fMRI data in neuroimaging research [18]. (Line 207 – 214)

Item 14:

Page 19: The use of one-way ANOVA to analyse paired data (MRI ALFF and FC) is highly inappropriate. This must be changed. 

Reply:

For ALFF and FC maps, paired T-test will be used for intra-group comparison and one-way ANOVA analysis will be used for inter-group comparison in the three stimulating site subgroups. The detailed information was shown in the statistical analyses part. (Line 371 – 400; Line 409 – 415)

Item 15:

Page 19: The level of significance must be stated. 

Reply:

The multiple comparison correction will be performed using Gaussian random field theory (GRF) with voxel p < 0.005, cluster p < 0.05. The Pearson correlation analyses were used to calculate the correlations between the changes in ALFF and FC with the clinical symptom scores at baseline and the end of the rTMS treatment (p < 0.05). (Line 415 – 427)

Item 16:

*IMPORTANT* There is no assessment of safety. This is a major oversight and needs to be added. This is non-negotiable. 

Reply:

We added the adverse events of rTMS treatment as the safety assessment at the end of outcomes part. (Line 307; Line 401 – 405)

Item 17:

*IMPORTANT* Why is there no sham group? 

Reply:

The sham group will not be designed in the present study, because the objective of this study is to compare the effect difference in different parameter combination (stimulating site, frequency, intensity and pulse number) of rTMS treating to PSCI and try to find out the optimal parameter combination, but not to confirm whether rTMS is effective for improving the cognitive function in patients with PSCI. Hence, an orthogonally designed study will be conducted.

---

## [Decision Letter · Decision Letter 1]

28 Jun 2022

The optimum parameters and neuroimaging mechanism of repetitive transcranial magnetic stimulation to post-stroke cognitive impairment, a protocol of an orthogonally-designed randomized controlled trial

PONE-D-21-30462R1

Dear Dr. LI,

We’re pleased to inform you that your manuscript has been judged scientifically suitable for publication and will be formally accepted for publication once it meets all outstanding technical requirements.

Kind regards,

Giuseppe Lanza, M.D., Ph.D.

Academic Editor

PLOS ONE

Additional Editor Comments (optional):

Reviewers' comments:

Reviewer's Responses to Questions

**Comments to the Author**

1. Does the manuscript provide a valid rationale for the proposed study, with clearly identified and justified research questions?

Reviewer #2: Yes

2. Is the protocol technically sound and planned in a manner that will lead to a meaningful outcome and allow testing the stated hypotheses?

Reviewer #2: Yes

3. Is the methodology feasible and described in sufficient detail to allow the work to be replicable?

Reviewer #2: Yes

4. Have the authors described where all data underlying the findings will be made available when the study is complete?

Reviewer #2: Yes

5. Is the manuscript presented in an intelligible fashion and written in standard English?

Reviewer #2: Yes

6. Review Comments to the Author

You may also provide optional suggestions and comments to authors that they might find helpful in planning their study.

Reviewer #2: I am satisfied that the authors have addressed all the issues previously identified. I am happy to support the acceptance of this manuscript.

7. PLOS authors have the option to publish the peer review history of their article (what does this mean?). If published, this will include your full peer review and any attached files.

Reviewer #2: No

---

## [Editor Report · Acceptance letter]

12 Jul 2022

PONE-D-21-30462R1 

The optimum parameters and neuroimaging mechanism of repetitive transcranial magnetic stimulation to post-stroke cognitive impairment, a protocol of an orthogonally-designed randomized controlled trial 

Dear Dr. LI:

I'm pleased to inform you that your manuscript has been deemed suitable for publication in PLOS ONE. Congratulations! Your manuscript is now with our production department. 

Kind regards, 

on behalf of

Dr. Giuseppe Lanza 

Academic Editor

PLOS ONE